# *Ehrlichia chaffeensis* Etf-3 Induces Host *RAB15* Upregulation for Bacterial Intracellular Growth

**DOI:** 10.3390/ijms25052551

**Published:** 2024-02-22

**Authors:** Nan Yang, Meifang Li, Shanhua Qin, Nan Duan, Xiaoxiao Li, Yuhong Zhou, Mengyao Wang, Yongxin Jin, Weihui Wu, Zhihui Cheng

**Affiliations:** 1The Key Laboratory of Molecular Microbiology and Technology, Ministry of Education, Nankai University, Tianjin 300071, China; 2120211170@mail.nankai.edu.cn (N.Y.); 2120211156@mail.nankai.edu.cn (M.L.); qinshanhua2000@163.com (S.Q.); duannan@mail.nankai.edu.cn (N.D.); 15230183753@126.com (X.L.); 2120231471@mail.nankai.edu.cn (Y.Z.); 2120231469@mail.nankai.edu.cn (M.W.); yxjin@nankai.edu.cn (Y.J.); wuweihui@nankai.edu.cn (W.W.); 2Department of Microbiology, College of Life Sciences, Nankai University, Tianjin 300071, China

**Keywords:** *Ehrlichia chaffeensis*, Etf-3, intracellular vesicle trafficking, RAB15, type IV secretion system effector

## Abstract

*Ehrlichia chaffeensis* infects human monocytes or macrophages and causes human monocytic ehrlichiosis (HME), an emerging life-threatening zoonosis. After internalization, *E. chaffeensis* resides in membrane-bound inclusions, *E. chaffeensis*-containing vesicles (ECVs), which have early endosome-like characteristics and fuse with early autophagosomes but not lysosomes, to evade host innate immune microbicidal mechanisms and obtain nutrients for bacterial intracellular growth. The mechanisms exploited by *E. chaffeensis* to modulate intracellular vesicle trafficking in host cells have not been comprehensively studied. Here, we demonstrate that *E. chaffeensis* type IV secretion system (T4SS) effector Etf-3 induces *RAB15* upregulation in host cells and that RAB15, which is localized on ECVs, inhibits ECV fusion with lysosomes and induces autophagy. We found that *E. chaffeensis* infection upregulated *RAB15* expression using qRT-PCR, and RAB15 was colocalized with *E. chaffeensis* using confocal microscopy. Silence of RAB15 using siRNA enhanced ECV maturation to late endosomes and fusion with lysosomes, as well as inhibited host cell autophagy. Overexpression of Etf-3 in host cells specifically induced *RAB15* upregulation and autophagy. Our findings deepen the understanding of *E. chaffeensis* pathogenesis and adaptation in hosts as well as the function of RAB15 and facilitate the development of new therapeutics for HME.

## 1. Introduction

*Ehrlichia chaffeensis*, a small tick-borne obligatory intracellular bacterium, infects human monocytes or macrophages and causes human monocytic ehrlichiosis (HME) [1]. HME is an emerging life-threatening zoonosis, which is mainly prevalent in the USA and has also been reported in Asia, Europe, Africa, and South America [1]. HME is generally characterized by headache, chills, acute fever, myalgia, and anorexia, accompanied by leukopenia, anemia, and elevation of serum hepatic aminotransferases [2]. Approximately 40–60% of patients require hospitalization, and the estimated case fatality rate for HME is approximately 3% [2]. The drug for HME treatment is doxycycline; however, this zoonosis is a particular threat to elderly and immunocompromised individuals [3]. Although a specific nanobody can block *E. chaffeensis* infection in cell culture and in a mouse model [4], vaccines are not available for HME. The illustration of *E. chaffeensis* intracellularly adaptative mechanisms has become a central focus for understanding the pathogenesis of HME and the development of new therapeutics.

After internalization of monocytes or macrophages, which are primary immune cells, *E. chaffeensis* resides in specialized membrane-bound inclusions, *E. chaffeensis*-containing vesicles (ECVs), which provide sanctuary for the evasion of host innate immune microbicidal mechanisms [5]. ECVs have early endosome-like characteristics, including transferrin (Tf), transferrin receptor (TfR), vacuolar-type HC-ATPase, the small GTPase RAB5, its effector early endosome antigen 1 (EEA1), and missing lysosome markers [6,7]. During *E. chaffeensis* intracellular growth, ECVs fuse with early autophagosomes to obtain host cell-derived nutrients. *Ehrlichia chaffeensis* exploits the type IV secretion system (T4SS) to secrete effectors into the host cell cytoplasm to modulate and hijack intracellular vesicle trafficking [8]. The effector Etf-1 enters mitochondria and inhibits the mitochondria-mediated apoptosis of host cells [9]. Etf-1 also induces autophagy through RAB5 and class III phosphatidylinositol 3-kinase to liberate catabolites for bacterial growth inside host cells [10]. Etf-2 competes with the RAB5 GTPase-activating protein for binding to RAB5-GTP on the surface of ECVs, which blocks GTP hydrolysis and consequently prevents the fusion of ECVs with lysosomes [11]. Etf-3 binds the ferritin light chain to induce ferritinophagy to obtain intracellular iron [12]. Future studies on the function of these effectors and their interaction with host cell components will advance our understanding of the molecular mechanisms of *E. chaffeensis* intracellular growth and the vesicle trafficking of host cells.

RAB GTPases are a large family of monomeric proteins that regulate intracellular vesicle trafficking [13]. The intracellular distribution of each RAB is consistent with their function in regulating vesicle trafficking [13]. RAB15 has been reported to colocalize with the early endosome markers RAB5 and TfR but not with the lysosome markers mannose 6-phosphate receptor (M6PR) [14]. RAB15 is involved in the regulation of endocytosis and early endosome fusion [15,16]. However, whether RAB15 is localized on ECVs or plays roles in *E. chaffeensis* intracellular growth remains to be illustrated. In this study, to gain insights into the RAB15′s function, we performed an immunofluorescent assay to identify its distribution and silenced RAB15 with siRNA to determine its roles in *E. chaffeensis* intracellular growth. Then, we overexpressed T4SS effectors in host cells and found that Etf-3 induced *RAB15* upregulation to facilitate *E. chaffeensis* intracellular growth.

## 2. Results

### 2.1. RAB15 Is Localized on the ECV Membrane and Upregulated upon E. chaffeensis Infection

RABs play important regulatory roles in intracellular vesicle trafficking. Since RAB15 is colocalized with RAB5 and TfR on the early endosomal membrane [14], and RAB5 and TfR are localized on the ECV membrane, we first investigated whether RAB15 is also localized on the ECV membrane. Confocal immunofluorescence microscopy results demonstrated that RAB15 was colocalized with *E. chaffeensis* in infected THP-1 cells (Figure 1a), indicating that RAB15 is localized on the ECV membrane. We then examined *RAB15* expression in *E. chaffeensis*-infected THP-1 cells and found that *RAB15* was upregulated in *E. chaffeensis*-infected THP-1 cells compared to uninfected cells (Figure 1b), suggesting that RAB15 may play important roles in *E. chaffeensis* intracellular growth.

### 2.2. RAB15 Is Critical for E. chaffeensis Intracellular Growth

Next, we investigated whether RAB15 is critical for *E. chaffeensis* intracellular growth. We silenced RAB15 in THP-1 cells and examined the intracellular growth of *E. chaffeensis*. THP-1 cells were transfected with siRNA targeting *RAB15* or control siRNA for 24 h and then infected with isolated *E. chaffeensis*. The intracellular growth of *E. chaffeensis* was determined with qRT-PCR. The results showed that the silence of RAB15 significantly reduced *E. chaffeensis* intracellular growth (Figure 2a). The silence efficiency of RAB15 was determined using Western blotting and qRT-PCR (Figure 2b and Appendix A).

### 2.3. RAB15 Inhibits ECV Maturation and Fusion with Lysosomes

Since ECVs do not maturate to late endosomes nor fuse with lysosomes, we then investigated whether RAB15 plays a role in these processes. RAB7 is localized on the membrane of late endosomes and lysosomes [13,17], and M6PR is localized on lysosomes [18]. We silenced RAB15 in THP-1 cells and examined the colocalization of RAB7 or M6PR with *E. chaffeensis*. Immunofluorescence microscopy results showed that the colocalization of RAB7 or M6PR with *E. chaffeensis* increased significantly in RAB15-silenced THP-1 cells compared to control cells (Figure 3a–d). These results indicate that RAB15 inhibits ECV maturation and fusion with lysosomes.

### 2.4. RAB15 Induces Autophagy in E. chaffeensis-Infected THP-1 Cells

After the internalization of host cells, *E. chaffeensis* induces autophagy to obtain nutrients for its intracellular growth [10]. We next investigated whether RAB15 is involved in autophagy induction. The conversion of soluble LC3-I to lipid-bound LC3-II is associated with the formation of autophagosomes [19]. In line with the previous report [10], LC3-II in infected THP-1 cells increased significantly compared to uninfected cells (Appendix A). We then infected RAB15-silenced THP-1 cells with *E. chaffeensis* and found that the increase in LC3-II induced by *E. chaffeensis* infection was abolished (Figure 4a). Taking into consideration the above, all these results indicate that *E. chaffeensis* induces autophagy in THP-1 cells for its growth via the upregulation of *RAB15*.

To confirm this, we treated RAB15-silenced THP-1 cells with rapamycin, which can induce autophagy [10], and then infected these cells with *E. chaffeensis*. As expected, the treatment of rapamycin induced autophagy in THP-1 cells and enhanced *E. chaffeensis* intracellular growth (Figure 4b,c), while silence of RAB15 reduced both the induction and the enhancement (Figure 4b,c).

### 2.5. Ehrlichia chaffeensis Etf-3 Induces RAB15 Upregulation in THP-1 Cells

Next, we investigated the mechanisms by which *E. chaffeensis* upregulates *RAB15* expression. Since *E. chaffeensis* translocates effectors through the T4SS to modulate host cell intracellular trafficking [8], we hypothesized that *RAB15* upregulation might be induced by the T4SS effectors. Then, we transfected THP-1 cells with pcDNA3.1 overexpressing the three T4SS effectors, Etf-1, Etf-2, or Etf-3 for 48 h, respectively (Appendix A). We found that only overexpression of Etf-3 induced *RAB15* upregulation in THP-1 cells (Figure 5a).

Next, we used peptide nucleic acid (PNA) to knock down Etf-3. PNA is a DNA mimic that has been shown to bind single- and double-stranded DNA and RNA with high affinity and specificity [20], and it inhibits transcription from double-stranded DNA [21] and translation from RNA [22]. Therefore, we used PNA that specifically binds near the translation start site of *etf-3* [12]. In line with the previous report [12], transfection of host cell-free *E. chaffeensis* with Etf-3 PNA significantly reduced the *etf-3* mRNA level and bacterial intracellular growth (Appendix A). The transfection with Etf-3 PNA significantly reduced the expression of *RAB15* in THP-1 cells (Figure 5b), confirming that Etf-3 induced *RAB15* upregulation.

Since overexpression of Etf-3 enhances *E. chaffeensis* intracellular growth [12], we examined whether the enhancement was partially due to increased autophagy induced by Etf-3 via RAB15. We found that LC3-II increased in Etf-3-overexpressing THP-1 cells (Figure 5c). All these results indicate that *E. chaffeensis* translocates Etf-3 into the host cell cytoplasm to induce the upregulation of *RAB15*, which facilitates bacterial intracellular growth.

## 3. Discussion

In this study, we found that *E. chaffeensis* infection upregulated the expression of *RAB15*. RAB15 is colocalized on the ECV membrane with RAB5 and inhibits ECV maturation to late endosomes. RABs regulate various intracellular traffic pathways in eukaryotic cells: endocytosis; secretion; and autophagy [23]. RABs are localized on the membrane of specific vesicles and recruit their downstream effectors to organize membrane microdomains that mediate vesicular trafficking [24]. RABs cycle between an inactive GDP-bound and an active GTP-bound state, and only at the GTP-bound state RABs bind to effectors and mediate downstream functions [24]. RAB5, which is recruited rapidly and transiently to early endosomes, is essential for the recruitment of RAB7 and the vesicle maturation progression [25]. During infection, *E. chaffeensis* blocks GTP hydrolysis to activate RAB5 on the ECV membrane but inhibits ECV maturation to late endosomes [3,11]. RAB15 counteracts the stimulatory effects of RAB5 on early endocytosis directly by sequestering RAB5 effectors or indirectly through undetermined interactions [15,16]. It is highly likely that RAB15 inhibits RAB7 recruitment to ECVs by the interaction with RAB5, which then inhibits ECV maturation to late endosomes.

We found that silence of RAB15 reduced autophagy in infected THP-1 cells. This result indicates that RAB15 participates in autophagy induction. Autophagy is a highly conserved intracellular degradation process during which cytoplasmic compartments are enclosed within double-membrane vesicles, which eventually fuse with lysosomes to degrade the enclosed contents [26]. Many RAB proteins are involved in various stages of autophagy. It has been reported that RAB1, RAB5, RAB7, RAB8, RAB9, RAB11, RAB23, RAB24, RAB25, RAB32, and RAB33B have key roles in canonical and non-canonical autophagy [26]. RABs interact with downstream RABs in cascades or bind effector proteins to regulate intracellular vesicle trafficking [13]. RAB15 may interact with other RABs or undermined effectors to induce autophagy indirectly or directly.

In host cells, *E. chaffeensis* exploits Etf-1 and Etf-2 to activate RAB5, while the activation of RAB5 leads to RAB7 recruitment, which can lead to ECV maturation to late endosomes and fusion with lysosomes [10,11,25]. *Ehrlichia chaffeensis* has to develop a mechanism to balance RAB5 activation to prevent RAB7 recruitment. *Legionella pneumophila* and *Coxiella burnetii* use the T4SS to secrete effectors into host cells, which interfere with vesicle trafficking, ubiquitylation, gene expression, and lipid metabolism to promote pathogen survival [27]. During evolution, *E. chaffeensis* cleverly and specifically secretes Etf-3 to induce the upregulation of *RAB15*, which can interfere with RAB5 activation. Another benefit is that RAB15 induces autophagy at the same time, which provides nutrients for *E. chaffeensis* intracellular growth. Since overexpression of Etf-1 and Etf-2 has no effect on *RAB15* expression, the upregulation of *RAB15* is not due to the recognition of bacterial proteins by the host cell’s innate immune system. The regulation of *RAB15* expression is largely unknown. Identifying all the changes in host gene expression caused by Etf-3 might provide candidates for regulating *RAB15* expression, which is under our investigation. Elucidating the mechanism by which Etf-3 induces *RAB15* upregulation will deepen our understanding of not only the interaction between pathogenic bacteria and host cells but also the regulation of intracellular vesicle trafficking.

New and efficient treatments for *E. chaffeensis* infection are in high demand. Due to the reduction in the bacterial genome during evolution, *E. chaffeensis* has only a few T4SS effectors in its genome, which results in multiple functions merging into each effector [8]. The function of Etf-1 includes apoptosis inhibition and autophagy induction in host cells, which are required for *E. chaffeensis* intracellular survival and growth [9,10]; thus, Etf-1 becomes a potential target for the development of new therapeutics. A nanobody blocking Etf-1 function can block *E. chaffeensis* infection in cell culture and in a mouse model [4]. Etf-3 is essential for *E. chaffeensis* to obtain intracellular iron [12]. We found that Etf-3 induces the upregulation of *RAB15*. RAB15 subsequently inhibits ECV maturation and fusion with lysosomes and induces autophagy, which facilitates *E. chaffeensis* in establishing a haven to evade host immune responses and acquire nutrients for intracellular growth. Thus, nanobodies targeting Etf-3 or chemicals reducing *RAB15* expression during *E. chaffeensis* infection will effectively reduce bacterial intracellular growth by blocking nutrient acquisition as well as the bacterial amount by causing ECVs to fuse with lysosomes. Our results shed light on treatments not only for HME but also for other diseases caused by intracellular bacterial infections.

## 4. Materials and Methods

### 4.1. Bacteria and Cell Culture

*Ehrlichia chaffeensis* Arkansas strain was propagated in THP-1 cells in RPMI 1640 medium supplemented with 2 mM _L_-glutamine and 10% fetal bovine serum (FBS) (Every Green, Huzhou, China) at 37 °C in 5% CO_2_ and 95% air, as described previously [28].

*Escherichia coli* strains (Appendix A) DH5α, used for general cloning, were cultured in LB broth supplemented with ampicillin (100 µg/mL), as necessary.

### 4.2. DNA Plasmids, Primers, and siRNAs

The plasmids and primers used in this study are listed in Appendix A. For small interfering RNA (siRNA) experiments, siRNA targeting human *RAB15*: 5′-GGCAUGGACUUCUAUGAAACAAG-3′ [29] and the control siRNA were obtained from RiBoBio (Guangzhou, China).

### 4.3. Transfection

To overexpress proteins, 6 × 10^5^ THP-1 cells were transfected with 2.5 µg pcDNA3.1, overexpressing indicated proteins using Lipofectamine 3000 (Invitrogen, Carlsbad, CA, USA) according to the manufacturer’s instructions and seeded in a 6-well plate. The cells were collected at 48 h after transfection.

To silence proteins, 6 × 10^5^ THP-1 cells were transfected with negative control siRNA or siRNA targeting *RAB15* using Lipofectamine RNAi MAX (Invitrogen) according to the manufacturer’s instructions and seeded in a 6-well plate. The cells were collected at 48 h after transfection.

### 4.4. Isolation of Host Cell-Free E. chaffeensis

*Ehrlichia chaffeensis* was isolated from infected THP-1 cells, as described previously [30]. Briefly, *E. chaffeensis*-infected THP-1 cells (2 × 10^7^ cells, >90% infected) were harvested at 600× *g* for 5 min at room temperature. The pellet was resuspended in 5 mL SPK buffer (200 mM sucrose, 50 mM potassium phosphate, pH 7.4), supplemented with 2 mM _L_-glutamine, and passed through a 23-gauge needle with a syringe on ice 30 times to rupture the host cell membrane. To remove unbroken cells and cell debris, the mixture was centrifuged at 1000× *g* for 5 min at 4 °C. Bacteria in the supernatant were collected using additional centrifugation at 10,000× *g* for 10 min at 4 °C. The bacterial pellet was suspended in a fresh culture medium for synchronous infection.

### 4.5. Quantitative RT-PCR

Total RNA was extracted from each sample and reverse transcribed to cDNA, as described previously [31]. The amounts of *E. chaffeensis* 16S rRNA, host cell *RAB15* mRNA, and *GAPDH* mRNA were determined with qRT-PCR using specific primers (Appendix A) and the ChamQ Universal SYBR qPCR Master Mix (Vazyme, Nanjing, China) on a StepOnePlus Real-Time PCR System (Applied Biosystems, Carlsbad, CA, USA). Relative bacterial number was determined as the amount of bacterial 16S rRNA normalized against that of human *GAPDH* mRNA.

### 4.6. Western Blotting

To detect the amount of RAB15 or LC3 in THP-1 cells, the same numbers of cells were collected using centrifugation at 600× *g* for 5 min at room temperature, resuspended in 1 × SDS sample buffer, and boiled for 10 min. The samples were then subjected to 12% SDS-polyacrylamide gel electrophoresis (SDS-PAGE) and transferred to a PVDF membrane (Millipore, Co., Cork, Ireland). The protein levels of RAB15, LC3, and β-Actin were determined using a rabbit polyclonal anti-RAB15 antibody (1:1000 dilution) (Abcam, Cambridge, UK, ab272636), a rabbit monoclonal anti-LC3B antibody (1:1000 dilution) (Abcam, ab192890), and a mouse monoclonal anti-β-Actin antibody (1:1000 dilution) (Invitrogen, MA5-15739), respectively.

### 4.7. Immunofluorescence Microscopy

Immunofluorescence microscopy was performed as previously described [32]. At 48 h p.i., *E. chaffeensis*-infected cells were cytospined onto slides, fixed with 4% paraformaldehyde (PFA) at room temperature for 10 min and then permeabilized and blocked with PGS (PBS supplemented with 0.5% bovine serum albumin, 0.1% gelatin and 0.1% saponin) at room temperature for 30 min. The fixed cells were incubated with anti-RAB15 (Abcam, ab272636), anti-RAB7 (Abcam, ab137029), or anti-M6PR (Abcam, ab124767) antibodies and anti-*E. chaffeensis* FtsZ antiserum, which were diluted at 1:100 in 5% bovine serum albumin (BSA) (Solarbio Life Sciences, Beijing, China) at 4 °C overnight. The cells were washed with PBS three times and then incubated with anti-rabbit IgG (EarthOx, San Francisco, CA, USA) and anti-mouse IgG (Cell Signaling Technology, Boston, MA, USA) antibodies, which were diluted at 1:400 in 5% BSA at room temperature for 1 h. After being washed with PBS three times, the DNA in the host cell nucleus and *E. chaffeensis* were stained with DAPI diluted at 1:500 in 5% BSA at room temperature for 10 min. The cells were examined using an Olympus FV1200 confocal microscope (Olympus, Tokyo, Japan). Three slides were examined for each sample. A total of 100–150 bacteria were observed on each slide.

### 4.8. Peptide Nucleic Acid Transfection

Peptide nucleic acid transfection to knockdown Etf-3 in *E. chaffeensis* was performed as described previously [12]. An antisense PNA oligomer targeting 30–45 bp following the start codon of *etf-3* (Etf-3 PNA) 5′-TGCTGCTCTTTGTTGA-3′ and a control PNA (CTL PNA) 5′-GGCTCTATACAC-3′ were synthesized by TAHEPNA Biotechnologies Co., Ltd. (Hangzhou, China). Three micrograms of Etf-3 PNA or CTL PNA dissolved in nuclease-free water was mixed with 100 μL of host cell-free *E. chaffeensis* in 0.3 M sucrose and then incubated on ice for 15 min. Electroporation was conducted at 2000 V, 25 μF, and 400 Ω with a 10 ms pulse using a Gene Pulser Xcell™ electroporation system (Bio-Rad, Hercules, CA, USA) in a 2 mm electroporation cuvette (Bio-Rad). Then, the PNA-transfected *E. chaffeensis* was transferred to a T25 flask to infect 5 × 10^5^ THP-1 cells and incubated at 37 °C for 2 h with gentle shaking every 15 min to facilitate bacterial internalization. To detect the effect of Etf-3 PNA, the infected cells were harvested at 48 h p.i. The expression level of *etf-3* was examined with qRT-PCR.

### 4.9. Statistical Analysis

All experiments were performed at least three times. Statistical analyses were performed using GraphPad Prism 9.0. The statistical significance of a two-group comparison was assessed using Student’s *t*-test (two-tailed). A value of *p* < 0.05 was considered significant.

## 5. Conclusions

In summary, we found that *E. chaffeensis* infection induced *RAB15* upregulation in host cells, and RAB15, which is localized on ECVs, inhibited ECV fusion with lysosomes and induced autophagy. The upregulation of *RAB15* was induced specifically by the *E. chaffeensis* T4SS effector Etf-3. These findings provide new insights into *E. chaffeensis* pathogenesis and adaptation in hosts, as well as the function of RAB15, and facilitate the development of new therapeutics for HME.

## Figures and Tables

**Figure 1 ijms-25-02551-f001:**
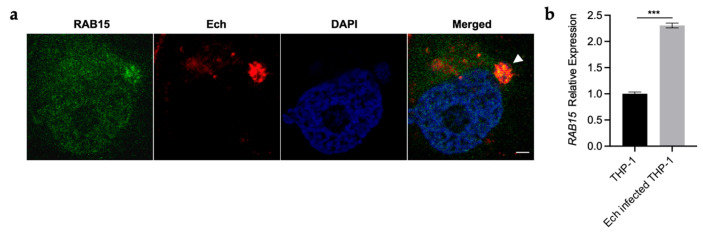
RAB15 is localized on the ECV membrane and upregulated upon *E. chaffeensis* infection. (**a**) RAB15 is colocalized with *E. chaffeensis* in infected THP-1 cells. At 48 h p.i., *E. chaffeensis*-infected THP-1 cells were cytospined onto slides and fixed with 4% PFA. *Ehrlichia chaffeensis* was labeled with mouse anti-*E. chaffeensis* FtsZ antiserum (red). RAB15 was labeled with rabbit anti-RAB15 antibody (green). The DNA in the host cell nucleus and *E. chaffeensis* were stained with DAPI (blue). The arrow indicates *E. chaffeensis* colocalized with RAB15. Scale bar, 5 µm. (**b**) The expression of *RAB15* is upregulated in *E. chaffeensis*-infected THP-1 cells. At 48 h p.i. 2 × 10^6^ uninfected or *E. chaffeensis*-infected THP-1 cells were collected, and total RNA was extracted from the cells. The expression level of *RAB15* was determined with qRT-PCR and normalized against that of human *GAPDH*. Relative values to the *RAB15* amount in uninfected THP-1 cells are shown. Data indicate means ± standard deviations (*n* = 3). The significant difference is represented by *p*-values determined with Student’s *t*-test (*** indicates *p* < 0.001).

**Figure 2 ijms-25-02551-f002:**
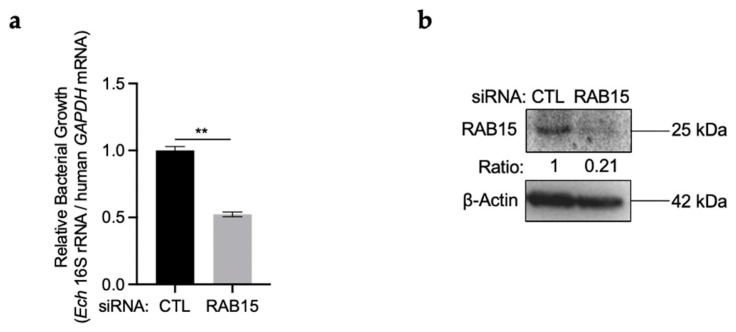
RAB15 is critical for *E. chaffeensis* intracellular growth. (**a**) Silence of RAB15 significantly reduces *E. chaffeensis* intracellular growth. THP-1 cells were transfected with siRNA targeting *RAB15* or control siRNA for 24 h and then infected with isolated *E. chaffeensis*. At 48 h p.i., total RNA was extracted from the cells. The intracellular growth of *E. chaffeensis* was determined with qRT-PCR and normalized against that of human *GAPDH*. Relative values to the *E. chaffeensis* 16S rRNA amount in control THP-1 cells are shown. Data indicate means ± standard deviations (*n* = 3). The significant difference is represented by *p*-values determined with Student’s *t*-test (** indicates *p* < 0.01). (**b**) The silence efficiency of RAB15 was determined using Western blotting. The numbers below the panels indicate the relative intensity of each protein band. The protein level of RAB15 in control cells is set as 1.

**Figure 3 ijms-25-02551-f003:**
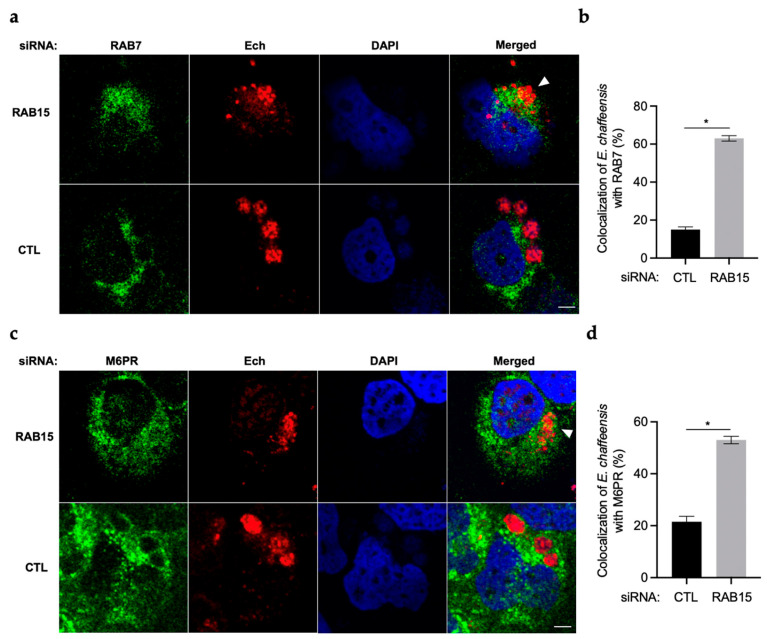
RAB15 inhibits ECV maturation and fusion with lysosomes. THP-1 cells were transfected with control siRNA or siRNA targeting *RAB15* for 24 h, then infected with isolated *E. chaffeensis*. At 48 h p.i., *E. chaffeensis*-infected THP-1 cells were cytospined onto slides and fixed with 4% PFA. *Ehrlichia chaffeensis* was labeled with mouse anti-*E. chaffeensis* FtsZ antiserum (red). RAB7 or M6PR was labeled with rabbit anti-RAB7 or anti-M6PR antibody (green). The DNA in the host cell nucleus and *E. chaffeensis* were stained with DAPI (blue). (**a**,**c**) Images show the colocalization of RAB7 or M6PR with *E. chaffeensis*. The arrow indicates *E. chaffeensis* colocalized with RAB7 or M6PR. Scale bar, 5 µm. (**b**,**d**) The numbers indicate the percentage of *E. chaffeensis* colocalized with RAB7 or M6PR relative to total intracellular bacteria (*n* = 3 slides). The significant differences are represented by *p*-values determined with Student’s *t*-test (* indicates *p* < 0.05).

**Figure 4 ijms-25-02551-f004:**
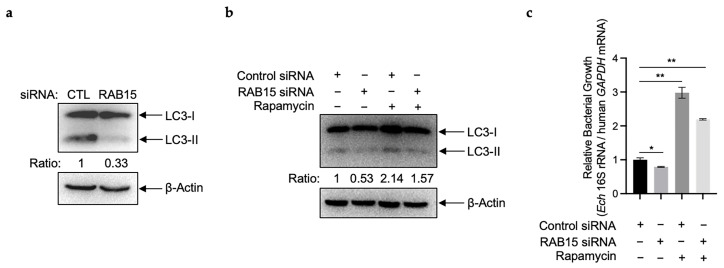
RAB15 induces autophagy in *E. chaffeensis*-infected THP-1 cells. (**a**) Silence of RAB15 causes the inhibition of autophagy. THP-1 cells were transfected with siRNA targeting *RAB15* or control siRNA for 24 h and then infected with isolated *E. chaffeensis*. At 48 h p.i., the expression of LC3 was determined using Western blotting. The numbers below the panels indicate the relative intensity of each protein band. The protein level of LC3-II in control cells is set as 1. (**b**) Silence of RAB15 reduces autophagy induced by rapamycin. THP-1 cells were transfected with siRNA targeting *RAB15* or control siRNA for 24 h. Then, the cells were pretreated with 0.1 µM rapamycin or 0.1% DMSO (control) for 2.5 h and infected with isolated *E. chaffeensis* in the continued presence of rapamycin or DMSO. At 48 h p.i., the expression of LC3 was determined using Western blotting. The numbers below the panels indicate the relative intensity of each protein band. The protein level of LC3-II in control cells is set as 1. (**c**) Silence of RAB15 reduces the enhancement of *E. chaffeensis* intracellular growth induced by rapamycin. THP-1 cells were transfected with siRNA targeting *RAB15* or control siRNA for 24 h. Then, the cells were pretreated with 0.1 µM rapamycin or 0.1% DMSO (control) for 2.5 h and infected with isolated *E. chaffeensis* in the continued presence of rapamycin or DMSO. At 48 h p.i., total RNA was extracted from the cells. The intracellular growth of *E. chaffeensis* was determined with qRT-PCR and normalized against that of human *GAPDH*. Relative values to the *E. chaffeensis* 16S rRNA amount in control THP-1 cells are shown. Data indicate means ± standard deviations (*n* = 3). The significant difference is represented by *p*-values determined with Student’s *t*-test (* indicates *p* < 0.05, and ** indicates *p* < 0.01).

**Figure 5 ijms-25-02551-f005:**
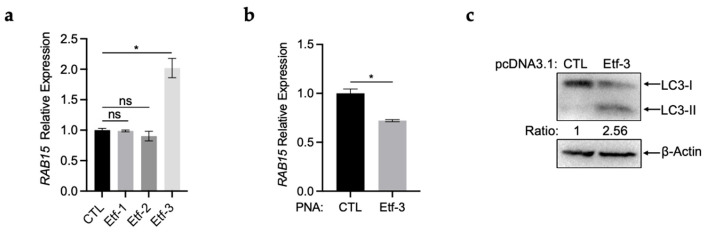
*Ehrlichia chaffeensis* Etf-3 induces *RAB15* upregulation in THP-1 cells. (**a**) *RAB15* is upregulated in THP-1 cells overexpressing *E. chaffeensis* Etf-3. THP-1 cells were transfected with pcDNA3.1 overexpressing Etf-1, Etf-2, or Etf-3, respectively, for 48 h. The expression of *RAB15* was determined with qRT-PCR and normalized against that of human *GAPDH*. Relative values to the *RAB15* amount in control cells are shown. Data indicate means ± standard deviations (*n* = 3). The significant differences are represented by *p*-values determined with Student’s *t*-test (ns indicates *p* > 0.05, and * indicates *p* < 0.05). (**b**) Transfection of Etf-3 PNA significantly reduces the expression of *RAB15* in THP-1 cells. THP-1 cells were synchronously infected with Etf-3 PNA-transfected *E. chaffeensis*. At 48 h p.i., total RNA was extracted from the cells. The expression level of *RAB15* was determined with qRT-PCR and normalized against that of human *GAPDH*. Relative values to the amount of CTL PNA transfected *E. chaffeensis* are shown. Data indicate means ± standard deviations (*n* = 3). The significant differences are represented by *p*-values determined with Student’s *t*-test (* indicates *p* < 0.05). (**c**) Overexpression of Etf-3 induces autophagy. THP-1 cells were transfected with pcDNA3.1 overexpressing Etf-3 for 48 h. The expression of LC3 was determined using Western blotting. The numbers below the panels indicate the relative intensity of each protein band. The protein level of LC3-II in control cells is set as 1.

## Data Availability

Data are contained within the article and Appendix A.

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
