# Peer review of "Ehrlichia chaffeensis Etf-3 Induces Host RAB15 Upregulation for Bacterial Intracellular Growth"

_ijms, 2024, doi:10.3390/ijms25052551_

Round 1

Reviewer 1 Report

Comments and Suggestions for Authors

This is an interesting manuscript about an important zoonotic pathogen. The authors address the mechanisms used by Ehrlichia chaffeensis to modulate intracellular vesicle trafficking in host cells, which can iinfluence treatment efficacy. Under this reviewer’s criteria, the manuscript is well written, nevertheless requiring some adaptations in order to increase its value for the readers.

Line 14 – replace with: one life-threatening emerging zoonosis

Lines 16-17 – insert commas: with early autophagosomes, but not lysosomes, to evade

Keywords – display alphabetically

Line 34 – not correct: … infectious zoonosis all over the world… – all zoonoses are infectious (so, delete infectious), and HME is prevalent mainly in the USA

Line 38 – insert semi-colon: doxycycline; however, this

Line 51, etc. – consider staring sentences with species names written out in full, i.e. Ehrlichia chaffeensis – the same for figures

The Discussion section could be slightly expanded

Please insert a Conclusion(s) section/subheading

References – these should be standardized, e.g. titles (lower case, whenever possible) and journal names (uppercase initials and abbreviated words)

Comments on the Quality of English Language

Minor editing of English language required

Author Response

Thank you very much for taking the time to review this manuscript. Please find the detailed responses below and the corrections highlighted in the re-submitted files.

Comments and Suggestions for Authors:

This is an interesting manuscript about an important zoonotic pathogen. The authors address the mechanisms used by Ehrlichia chaffeensis to modulate intracellular vesicle trafficking in host cells, which can influence treatment efficacy. Under this reviewer’s criteria, the manuscript is well written, nevertheless requiring some adaptations in order to increase its value for the readers.

Line 14 – replace with: one life-threatening emerging zoonosis

Response: According to the comment, we have modified the sentence (line 14).

Lines 16-17 – insert commas: with early autophagosomes, but not lysosomes, to evade

Response: According to the comment, we have modified the sentence (line 16-17).

Keywords – display alphabetically

Response: According to the comment, we have rearranged the keywords (line 28-29).

Line 34 – not correct: … infectious zoonosis all over the world… – all zoonoses are infectious (so, delete infectious), and HME is prevalent mainly in the USA

Response: According to the comment, we have modified the sentence (line 34-35).

Line 38 – insert semi-colon: doxycycline; however, this

Response: According to the comment, we have modified the sentence (line 39).

Line 51, etc. – consider staring sentences with species names written out in full, i.e. Ehrlichia chaffeensis – the same for figures

Response: According to the comment, we have modified the manuscript thoroughly.

The Discussion section could be slightly expanded

Response: According to the comment, we have expanded the discussion slightly.

Please insert a Conclusion(s) section/subheading

Response: According to the comment, we have added a conclusion section (line 349-355).

References – these should be standardized, e.g. titles (lower case, whenever possible) and journal names (uppercase initials and abbreviated words)

Response: According to the comment, we have standardized references.

Comments on the Quality of English Language:

Minor editing of English language required

Response: According to the comment, we have the manuscript edited by MDPI English Editing.

Reviewer 2 Report

Comments and Suggestions for Authors

The manuscript under title (Ehrlichia chaffeensis Etf-3 induces host RAB15 upregulation for 2 bacterial intracellular growth) is well designed, presented and constructed. Its findings have confirmed that E. chaffeensis infection upregulates the expression of 209 RAB15. RAB15 is colocalized on ECV membrane with RAB5 and inhibits ECV maturation 210 to late endosomes.

Few comments need clarification from authors

1. Where is the conclusion in the discussion section?

2. How these findings help in the therapeutic design for HME ?

Author Response

Thank you very much for taking the time to review this manuscript. Please find the detailed responses below and the corrections highlighted in the re-submitted files.

Comments and Suggestions for Authors:

The manuscript under title (Ehrlichia chaffeensis Etf-3 induces host RAB15 upregulation for bacterial intracellular growth) is well designed, presented and constructed. Its findings have confirmed that E. chaffeensis infection upregulates the expression of RAB15. RAB15 is colocalized on ECV membrane with RAB5 and inhibits ECV maturation to late endosomes.

Few comments need clarification from authors:

  1. Where is the conclusion in the discussion section?

Response: According to the comment, we have added a conclusion section (line 349-355).

  1. How these findings help in the therapeutic design for HME?

Response: According to the comment, we have discussed it in the Discussion section (line 258-266).

Reviewer 3 Report

Comments and Suggestions for Authors

The manuscript (ijms-2883249) describes Etf-3 induces host RAB15 upregulation for bacterial intracellular growth in Ehrlichia chaffeensis. In this study, E. chaffeensis type 4 secretion system effector Etf-3 induces RAB15 upregulation in host cells was demonstrate. It also indicated that RAB15 localized on ECVs inhibits ECV fusion with lysosomes and induces autophagy. The findings reported by the Authors of this manuscript are very important because it is deepening the understanding of E. chaffeensis pathogenesis and adaptation in hosts as well as the function of RAB15 and facilitate the development of new therapeutics for human monocytic erhlichiosis. Therefore, this manuscript falls into scopus of the journal and presents important data relevant to develope of novel therapeutics for human monocytic ehrlichiosis . Overall, the study design is well and the manuscript is also well-written.

Author Response

Comments and Suggestions for Authors:

The manuscript (ijms-2883249) describes Etf-3 induces host RAB15 upregulation for bacterial intracellular growth in Ehrlichia chaffeensis. In this study, E. chaffeensis type 4 secretion system effector Etf-3 induces RAB15 upregulation in host cells was demonstrate. It also indicated that RAB15 localized on ECVs inhibits ECV fusion with lysosomes and induces autophagy. The findings reported by the Authors of this manuscript are very important because it is deepening the understanding of E. chaffeensispathogenesis and adaptation in hosts as well as the function of RAB15 and facilitate the development of new therapeutics for human monocytic ehrlichiosis. Therefore, this manuscript falls into scopes of the journal and presents important data relevant to develop of novel therapeutics for human monocytic ehrlichiosis. Overall, the study design is well and the manuscript is also well-written.

Response: Thank you for the comments.

Reviewer 4 Report

Comments and Suggestions for Authors

The aim of this study was to investigate the mechanisms employed by Ehrlichia chaffeensis, a bacterium that causes human monocytic ehrlichiosis (HME), to modulate intracellular vesicle trafficking in host cells. The authors demonstrated that a type 4 secretion system (T4SS) effector called Etf-3 induced upregulation of the host protein RAB15. RAB15, localized on E. chaffeensis-containing vesicles (ECVs), inhibited their fusion with lysosomes and induced autophagy in host cells. The researchers observed that E. chaffeensis infection led to increased expression of RAB15, and silencing RAB15 enhanced ECV maturation and fusion with lysosomes while inhibiting host cell autophagy. Moreover, overexpression of Etf-3 induced RAB15 upregulation and autophagy. These findings deepen our understanding of E. chaffeensis pathogenesis and adaptation in hosts, shed light on the function of RAB15, and may contribute to the development of new therapeutics for HME.

After carefully examining the manuscript, I am pleased to affirm that the content is exceptionally well-articulated and elucidated. The introduction provides a comprehensive overview of the subject matter, while the materials and methods section offers clear insight into the study's procedures. The results and discussion are particularly commendable, being meticulously written and presented. Additionally, the inclusion of original images and supplementary materials further enhances the clarity and depth of the findings, making them visually and conceptually accessible. I extend my heartfelt congratulations to the authors for their exemplary work. Based on the quality of the manuscript, I recommend considering it for publication.

Author Response

Comments and Suggestions for Authors:

The aim of this study was to investigate the mechanisms employed by Ehrlichia chaffeensis, a bacterium that causes human monocytic ehrlichiosis (HME), to modulate intracellular vesicle trafficking in host cells. The authors demonstrated that a type 4 secretion system (T4SS) effector called Etf-3 induced upregulation of the host protein RAB15. RAB15, localized on E. chaffeensis-containing vesicles (ECVs), inhibited their fusion with lysosomes and induced autophagy in host cells. The researchers observed that E. chaffeensis infection led to increased expression of RAB15, and silencing RAB15 enhanced ECV maturation and fusion with lysosomes while inhibiting host cell autophagy. Moreover, overexpression of Etf-3 induced RAB15 upregulation and autophagy. These findings deepen our understanding of E. chaffeensis pathogenesis and adaptation in hosts, shed light on the function of RAB15, and may contribute to the development of new therapeutics for HME.

After carefully examining the manuscript, I am pleased to affirm that the content is exceptionally well-articulated and elucidated. The introduction provides a comprehensive overview of the subject matter, while the materials and methods section offers clear insight into the study's procedures. The results and discussion are particularly commendable, being meticulously written and presented. Additionally, the inclusion of original images and supplementary materials further enhances the clarity and depth of the findings, making them visually and conceptually accessible. I extend my heartfelt congratulations to the authors for their exemplary work. Based on the quality of the manuscript, I recommend considering it for publication.

Response: Thank you for the comments.